# How Training Data Affect the Accuracy and Robustness of Neural Networks for Image Classification

## Abstract

Recent work has demonstrated the lack of robustness of well-trained deep neural networks (DNNs) to adversarial examples. For example, visually indistinguishable perturbations, when mixed with an original image, can easily lead deep learning models to misclassifications. In light of a recent study on the mutual influence between robustness and accuracy over 18 different ImageNet models, this paper investigates how training data affect the accuracy and robustness of deep neural networks. We conduct extensive experiments on four different datasets, including CIFAR-10, MNIST, STL-10, and Tiny ImageNet, with several representative neural networks. Our results reveal previously unknown phenomena that exist between the size of training data and characteristics of the resulting models. In particular, besides confirming that the model accuracy improves as the amount of training data increases, we also observe that the model robustness improves initially, but there exists a turning point after which robustness starts to decrease. How and when such turning points occur vary for different neural networks and different datasets.

## 1 Introduction

Deep neural models have achieved ground-breaking results for a growing list of tasks such as image classification (Krizhevsky et al., 2012), speech recognition (Hinton et al., 2012), and the game of Go (Silver et al., 2016). Despite these accomplishments, research has discovered that existing deep neural networks are easily susceptible to various attacks. Szegedy et al. (2013) are the first to show the existence of *adversarial examples* in the image classification. Specifically, they demonstrate how to cause a network to misclassify an image by applying certain visually imperceptible perturbations. Clearly, this finding hinders the adoption of deep neural networks in practice, especially in safety-critical scenarios. Indeed, Evtimov et al. (2017) reveal that slight alterations to road signs can cause classifiers to predict a "STOP" sign as a "Speed Limit" sign.

Apart from image classification (Carlini & Wagner, 2016; Kurakin et al., 2016; Chen et al., 2017b), existing work has also investigated the robustness of deep learning models in other application domains, such as natural language processing (Jia & Liang, 2017; Cheng et al., 2018), image captioning (Chen et al., 2017a; Xu et al., 2018), and speech recognition (Carlini & Wagner, 2018). In a recent study, Su et al. (2018) empirically analyze how model accuracy and robustness interact on 18 deep image classification models, and provide several findings, the most interesting of which is that the sole pursuit of accuracy sacrifices robustness.

In this paper, we study how accuracy and robustness interact from a different perspective: the influence of training data on deep learning models. In particular, we aim at demystifying the relationship between the size of training data and important properties of the trained neural network, namely its accuracy and robustness. Our experiments are conducted with the CIFAR-10, MNIST, STL-10, and Tiny ImageNet datasets; and neural networks are chosen from a simple 2-layer perceptron to state-of-the-art DenseNet models. For each dataset, we split the training data into sub-datasets with strictly inclusion relationship (*i.e.*, having the current sub-dataset a *strict subset* of the next sub-dataset) and ensure all the sub-datasets are balanced (*i.e.*, having the same number of images for each label in the original dataset). We then train all targeted neural networks on these sub-datasets and investigate how their robustness and accuracy vary *w.r.t.* the size of the underlying training dataset. Our study reveals several insights, and we summarize the main contributions below:

- We present the first comprehensive study to show how model robustness changes with increased training data for five representative neural networks, including a 2-layer perceptron, multi-layer CNNs (with a similar structure as those of Alexnet and VGG), ResNet, and DenseNet, on four different datasets (CIFAR-10, MNIST, STL-10, and Tiny ImageNet).

- We demonstrate that the robustness of a simple linear regression model can decrease as the amount of training data increases via a closed-form calculation.

- We find that model accuracy continues to improve with increased training data. Similarly, model robustness also improves, but may start to deteriorate when training data continue to increase. The occurrence of turning points depends on the deep neural network as well as the dataset on which it is trained.

## 2 MOTIVATION

Increasing training data helps to increase the accuracy of models, however, the general trend in model robustness is relatively understudied in the literature. This section presents a simple example to illustrate that, with more training data, a model can be more accurate but less robust. To allow a closed-form calculation, and simple and clear presentation, we consider the linear regression model.

First, we generate a set of data points in the form of $(x, y)$, where $x$ is the input, $y$ the output and $y = a * x + \mu$. Here in the example, we firstly draw a constant $a$ from $(-10, 10)$, then each pair of $(x, y)$ is generated by sampling $x$ from $(0, 10)$ and $\mu$ from $(-1, 1)$. Then, we split the generated dataset into training data and testing data randomly.

Given the generated dataset, we apply the closed-form formula to compute the optimal value for the coefficients of a linear regression model. The accuracy of the model $\mathcal{M}$ is obtained by calculating the mean squared error (MSE) from the testing set. Regarding the robustness of the model, we search for the minimum $|\delta|$ such that $|(\mathcal{M}(x_t) - y_t)/y_t| < \theta$ and $|(\mathcal{M}(x_t + \delta) - y_t)/y_t| > \theta$ holds, where $(x_t, y_t)$ denotes a data pair in the testing set. Essentially, we define an estimation to be correct if its relative error is within a bound, denoted by $\theta$. In other words, we look for the largest amount of distortion to which the model $\mathcal{M}$ is immune. Here, we set the bound of the relative error $\theta = 0.025$ and enumerate $\delta$ from 0 by a step of 0.0001.

Let us consider a concrete example with training sets $S_1$ and $S_2$, and testing set $T$:

$$
\begin{aligned}
S_1 &= \{(1.35, 9.52), (2.42, 16.7), (4.02, 28.03)\} \\
S_2 &= S_1 \cup \{(8.59, 60.22), (3.85, 25.74), (6.71, 47.2)\} \\
T &= \{(4.78, 33.24), (9.71, 67.78)\}
\end{aligned}
$$

We apply closed-form formula to compute two models $\mathcal{M}_1$ and $\mathcal{M}_2$ on $S_1$ and $S_2$ respectively. On testing set $T$, the MSE of $\mathcal{M}_1$ is 0.0450 and that of $\mathcal{M}_2$ is 0.0396, meaning that $\mathcal{M}_2$ is more accurate. Averaging the value of $\delta$ over the entire testing set $T$, we get 0.2194 for $\mathcal{M}_1$ and 0.2125 for $\mathcal{M}_2$, which indicates that $\mathcal{M}_1$ is more robust than $\mathcal{M}_2$.

To summarize, we show, in a linear regression model, as the volume of training data increases, the model can become more accurate but less robust. For the rest of this paper, we will study if this phenomenon is transferrable to deep neural networks in image classification.

## 3 BACKGROUND

This section introduces the components of our study, including the evaluated neural networks, the robustness property, and the datasets.

### 3.1 NEURAL NETWORKS AND NOTATION

Our study covers the following neural networks:

– **2-Layer Perceptron.** This is the simplest model with one hidden layer and a softmax output layer.

– **Simple CNN.** We use a 7-layer Alex-like (Krizhevsky et al., 2012) model for MNIST and CIFAR (with the same structure in Carlini & Wagner (2016)). It consists of four convolutional layers and three fully connected layers followed by the softmax output layer. The kernel sizes are all $3 \times 3$ for the convolutional layers. For STL-10 and Tiny ImageNet, we also use a deeper CNN (20 layers) with a similar structure as VGG (Krizhevsky et al., 2012).

– **ResNet.** ResNet was proposed to alleviate the vanishing gradient problem for training very deep neural networks (He et al., 2016). ResNet introduces a novel structure where each layer learns the residual functions with reference to the input by adding skip-layer paths, or "identity shortcut connections." In this paper, we use ResNet-32 for CIFAR-10 and ResNet-20 for MNIST, STL-10, Tiny ImageNet, respectively.

– **DenseNet.** This model was proposed by Huang et al. (2017) to further utilize "identity shortcut connections" across different layers. It connects all layers with each other within a dense block. In this paper, we study DenseNet with a depth of 40 for CIFAR-10, STL-10 and Tiny ImageNet.

## 3.2 ROBUSTNESS OF NEURAL NETWORKS

In this paper, we evaluate the robustness of models using state-of-the-art adversarial attacks. There are two kinds of adversarial attacks: *targeted* and *untargeted*. Given a valid input with label $t$, an untargeted attack searches for an input $x'$ such that the prediction on $x'$ is different from $t$, and $x'$ is very close to $x$. As for an targeted attack, if the target label is $t'$ ($t' \neq t$), it searches for an input $x'$ such that the prediction on $x'$ is $t'$ and $x'$ is close to $x$. To measure the distance between $x'$ and the original $x$, we use two widely adopted metrics — $\ell_2$ and $\ell_\infty$ — for adversarial perturbations (Carlini & Wagner, 2016; 2017; Chen et al., 2017b).

When evaluating the robustness of a model, the key is to measure the amount of distortions, when added to an image, cause the model's prediction to be incorrect regardless of predicted labels. Thus, in this paper, we focus on untargeted attacks of the following attack methods:

– **Iterative Fast Gradient Sign Mehtod (I-FGSM).** Kurakin *et al.* (Kurakin et al., 2016) proposed the I-FGSM attack to address the low success rate of the FGSM attack (Goodfellow et al., 2014) by applying FGSM multiple times with a finer distortion. One step of I-FGSM is:

$$x_{i+1} \leftarrow \text{clip}[x_i + \epsilon \, \text{sgn}(\nabla J(x_i, t))] \tag{1}$$

where $\text{sgn}(\nabla J(x_0, t))$ is the sign of the gradient of the training loss *w.r.t.* $x_i$, and $\text{clip}(x)$ is used to guarantee that the generated image $x$ is valid within the pixel range. This operation maximizes the training loss $J$ by updating $x$. The I-FGSM repeats this step multiple times, if the number of iterations is set to $T$, the per-iteration perturbation is set to $\frac{\epsilon}{T}\text{sgn}(\nabla J(x_0, t))$. It usually finds adversarial examples with small $\ell_\infty$ distortions. In this paper, we use the average $\ell_\infty$ distortions of adversarial images constructed by the I-FGSM attack as a measurement of the robustness of models.

– **Carlini and Wagner's attack (CW).** Carlini & Wagner (2016) formulates the problem of generating untargeted adversarial examples as the following optimization problem:

$$\begin{aligned} \min_x \quad & cf(x, t) + ||x - x_0||_2^2 \\ \text{s.t.} \quad & x \in [0, 1]^n \end{aligned} \tag{2}$$

where $n$ is the dimension of the images, and $f(x, t)$ is a loss function to determine if an attack succeeds for input $x$ with an original label $t$. In this work, the following loss function is used:

$$f(x, t) = \max\left\{ \text{Logit}(x)_t - \max_{i \neq t}\left[\text{Logit}(x)_i\right], -\kappa \right\} \tag{3}$$

where $\text{Logit}(x)$ denotes the vector representation of $x$ at the logit layer, and $\kappa$ denotes the confidence level. CW is by far one of the strongest attacks that construct adversarial images with small $\ell_2$ perturbations. In this paper, we use the average $\ell_2$ distortions of adversarial images constructed by this attack as another measurement of the robustness of models.

Table 1: Architecture and size of neural networks under robustness evaluation for each dataset

| Dataset | Name | Layers | Parameters |
|---------|------|--------|------------|
| CIFAR-10 | 2-Layer Perceptron | 2 | 3,157,002 |
| | Simple CNN | 7 | 1,147,978 |
| | ResNet | 32 | 467,946 |
| | DenseNet | 40 | 1,019,722 |
| MNIST | 2-Layer Perceptron | 2 | 814,090 |
| | Simple CNN | 7 | 312,202 |
| | ResNet | 20 | 272,778 |
| STL-10 | Simple CNN | 20 | 7,257,322 |
| | ResNet | 20 | 278,186 |
| | DenseNet | 40 | 1,019,722 |
| Tiny ImageNet | Simple CNN | 20 | 6,044,072 |
| | DenseNet | 40 | 1,105,032 |

## 3.3 DATASETS

We choose four representative datasets for our experiments: MNIST, CIFAR-10, STL-10[1] and Tiny ImageNet[2]. MNIST (LeCun et al., 1998) consists of 60,000 training data and 10,000 testing data for handwritten digit recognition. CIAFR-10 (Krizhevsky & Hinton, 2009) consists of 60,000 color images with 10 classes. STL-10 contains images with higher resolutions, which are acquired from labeled examples on ImageNet (Coates et al., 2011). It has 13,000 labeled data in total for 10 classes. In this paper, we divide the labeled data into 10,000 training data (1,000 per class) and 3,000 testing data (300 per class). Tiny ImageNet is a scaled-down version of ImageNet with 200 classes: each class has 500 training images, 50 validation images, and 50 test images. We use the validation images for accuracy and robustness evaluation since the labels of test images are not publicly available.

## 4 EXPERIMENTS

This section presents the details of our experiments, and summarizes and discusses our findings. Recall our goal is to reveal how training data affect the accuracy and robustness of neural networks.

## 4.1 SETUP

For each dataset in Section 3.3, we partition its training set into $n$ sub-datasets with a strict inclusion relationship: $S_1 \subset S_2 \subset ... \subset S_n$. Each sub-dataset contains the same number of images for each training label. For each neural network architecture $\mathcal{M}$ described in Section 3.1, we train a model on each of these sub-datasets to obtain $\mathcal{M}_1, ..., \mathcal{M}_n$. Finally, we record the accuracy and robustness of each $\mathcal{M}_i$. We use the same set of sub-datasets to train respective neural networks for each dataset.

For each dataset, we study several different neural networks shown in Table 1. On one hand, the neural networks are chosen based on their performance on the datasets. For example, 2-layer perceptron and 7-layer Simple CNN are used on CIFAR-10 and MNIST, but not STL-10 and Tiny ImageNet since they are generally considered to be beyond simple model architectures[3]. On the other hand, our choices include fully-connected network (2-layer perceptron), vanilla CNN (Simple CNN), CNN with residual connections (ResNet) and CNN with densely connected blocks (DenseNet) to ensure the sufficient diversity in neural network architectures.

---

[1]https://cs.stanford.edu/~acoates/stl10/

[2]https://tiny-imagenet.herokuapp.com/

[3]We do not apply DenseNet on MNIST due to its simplicity.

To evaluate the robustness of each model, we conduct the I-FGSM attack to obtain adversarial images in $\ell_\infty$ distortions and the CW attack to obtain those in $\ell_2$ distortions. We only consider original images that are correctly classified to avoid trial attacks when generating adversarial images. Given an attack on $\mathcal{M}_i$ with a valid input image $x$ of label $t$, we define the attack to be successful if $\mathcal{M}_i(x') \neq t$, where $x'$ is the adversarial image generated by the attack. The *attack success rate* is defined by the percentage of successful attacks. Both the CW and I-FGSM attacks can achieve 100% success rate on all models while the average distortions vary with different models and datasets. Equation (4) computes the robustness score of a model $\mathcal{M}_i$ where $X$ is the set of images that all $\mathcal{M}_i$ ($1 \leq i \leq n$) predict correctly.

$$Robustness(\mathcal{M}_i) = \frac{\sum_{x \in X} ||x - x'||_p}{|X|} \tag{4}$$

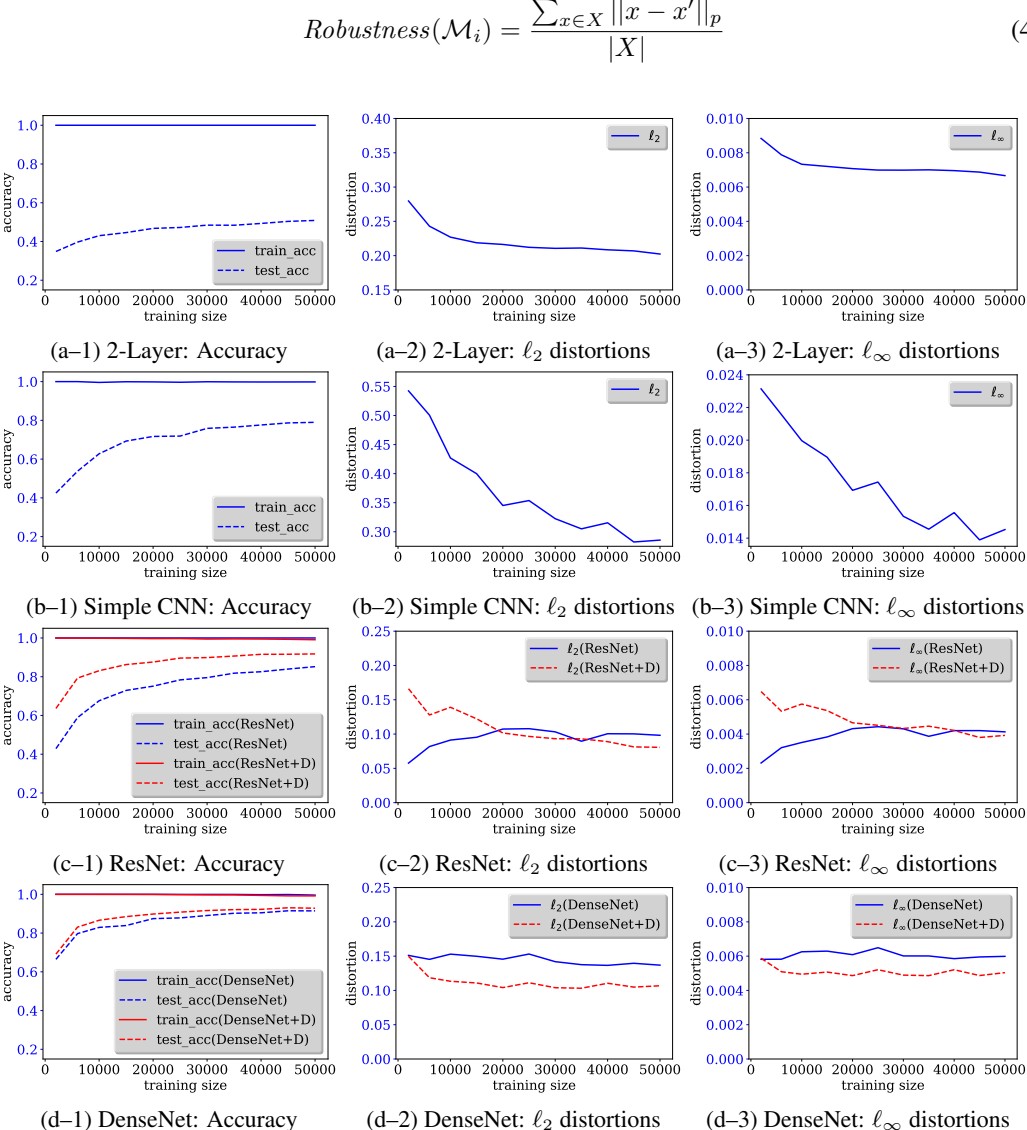

Figure 1: Results of accuracy and robustness of different neural networks on increasing training data on CIFAR-10. (ResNet/DenseNet+D represents ResNet/DenseNet models with data augmentation)

## 4.2 RELATIONSHIP BETWEEN SIZE OF TRAINING DATA, MODEL ACCURACY AND ROBUSTNESS

Results on CIFAR-10, MNIST, STL-10 and Tiny ImageNet are shown in Figs 1–4, respectively. The x-axis of each figure shows the size of sub-datasets, and the y-axis shows the accuracy or robustness of the models trained on each sub-dataset. The first column of each figure shows how the accuracy

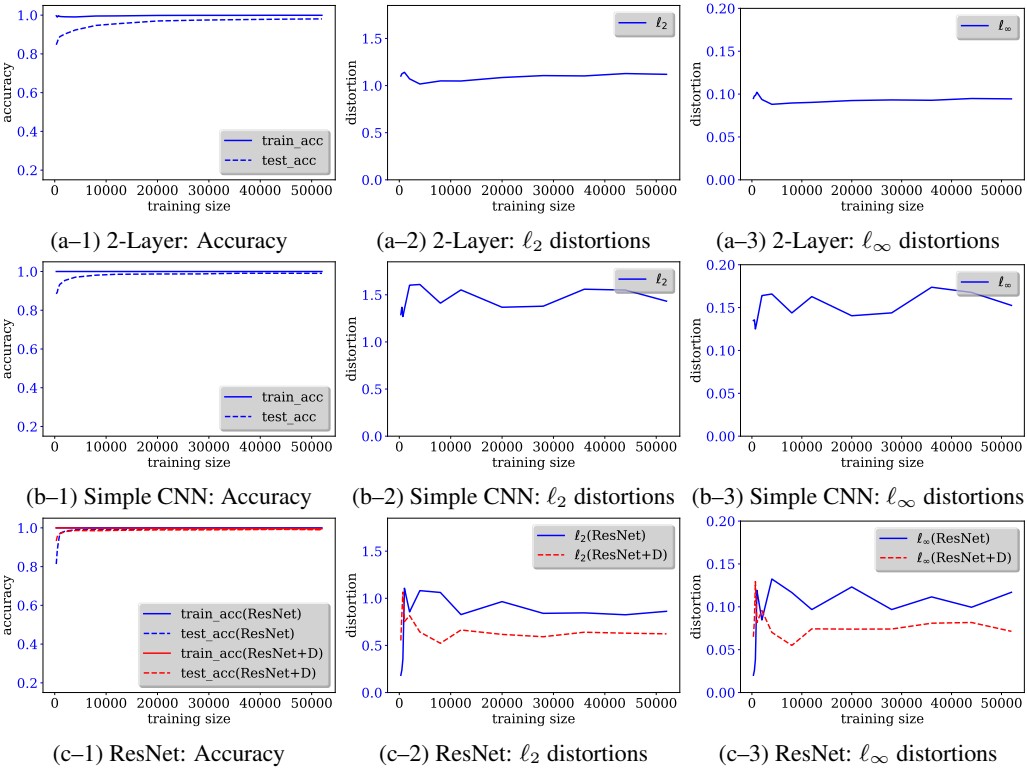

Figure 2: Results of accuracy and robustness of different neural networks on increasing training data on MNIST dataset.(ResNet+D represents ResNet models trained with data augmentation.)

of different neural networks changes with increased size of the training data. When training more complex models like ResNet and DenseNet, we consider both with and without data augmentation. Regarding data augmentation, we apply random shifting and flipping for ResNet; and random shifting, flipping and rotation for DenseNet.

The second and third column of each figure show how the robustness of models trained on increasingly larger training datasets changes measured by the $\ell_2$ distortions and $\ell_\infty$ distortions. For example, in Fig (a–2), the x-axis shows the size of the sub-datasets, and the y-axis shows the robustness of all 2-layer perceptron models (11 in total) trained with each of the sub-datasets. Their robustness is measured by the average $\ell_2$ distortions of the successful CW attacks on all 11 models. Similarly, the third column of Fig 1 shows how the robustness changes measured by the $\ell_\infty$ distortions of I-FGSM. On each dataset, we use the same scale to draw the robustness of different neural networks for an easy comparison across different neural networks. Note that the robustness of different models might reside in different ranges. For example, on CIFAR-10, 7-layer Simple CNN model is overall more robust than DenseNet ($\ell_2$ distortion in the range of 0.30 to 0.55, versus 0 to 0.25).

Summarizing results on all four datasets and different network architectures, we observe the following:

- For CIFAR-10 and STL-10 in Figs 1 and 3, we observe that, when the accuracy has reached a certain level, increasing the size of training dataset will continue to improve the accuracy but sacrifice the robustness. For CIFAR-10, ResNet (without data augmentation) and DenseNet (without data augmentation) are the only exception, where the robustness doesn't change much along with the accuracy. For STL-10, the only exception is ResNet (without data augmentation).

- In the case of Tiny ImageNet in Figure 4, both accuracy and robustness increase when a larger training dataset is used. It is worth noting that Tiny ImageNet only contains a small subset of ImageNet and is insufficient for obtaining good test accuracy. The model is still in the region of starving for training data. In this case, increasing the number of training data will benefit both the accuracy and robustness.

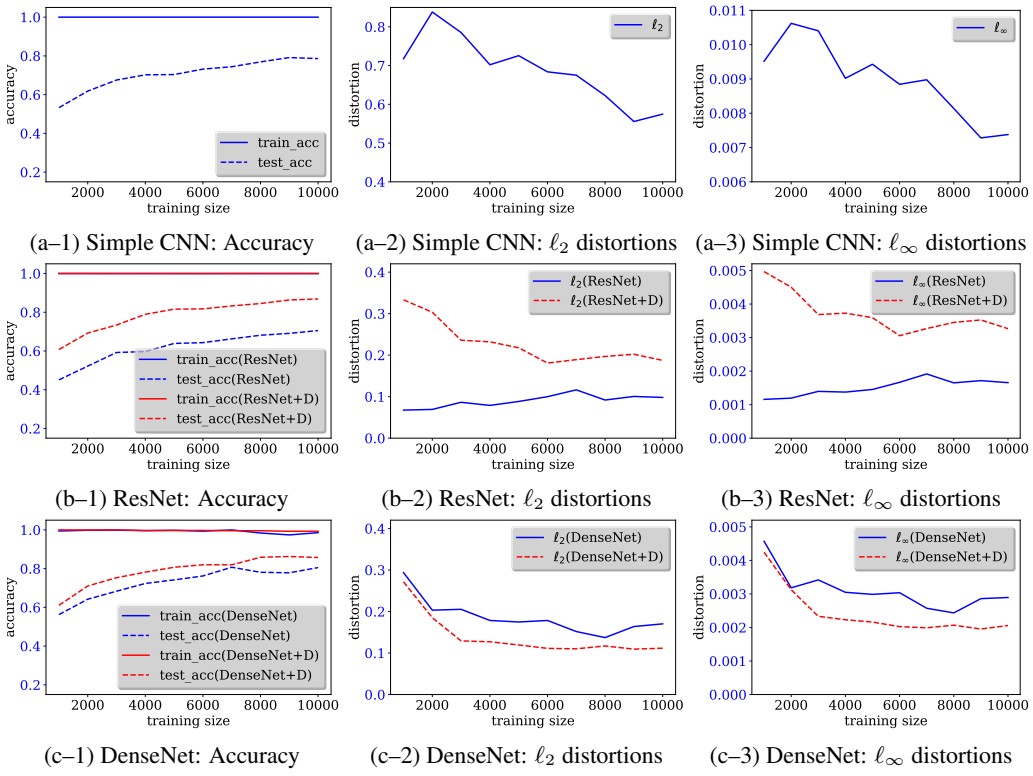

Figure 3: Results of accuracy and robustness of different neural networks on increasing training data on STL-10. (ResNet/DenseNet+D represents ResNet/DenseNet models with data augmentation.)

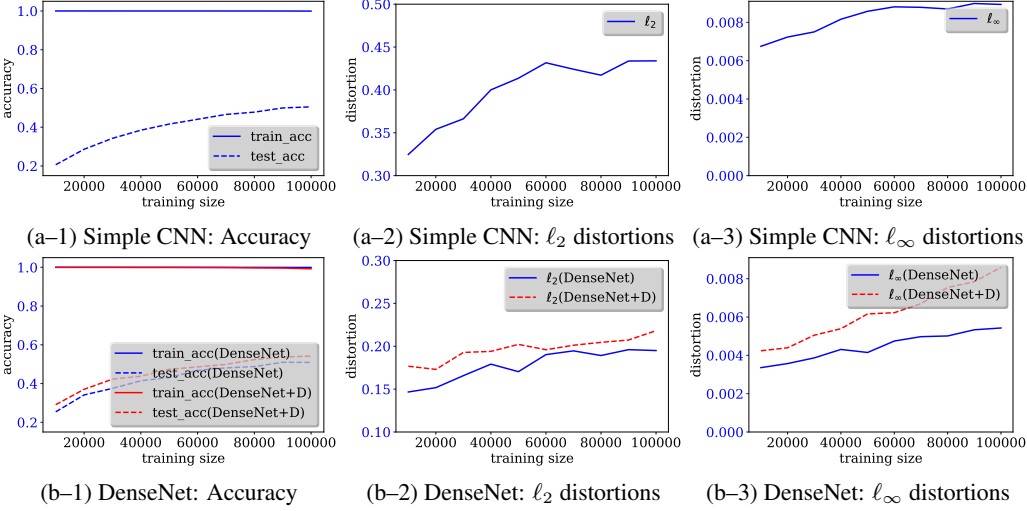

Figure 4: Results of accuracy and robustness of different neural networks on increasing training data on Tiny ImageNet dataset. (DenseNet+D represents DenseNet models trained with data augmentation.)

- For the MNIST dataset, test accuracy quickly increases to a very high level and then saturates. The robustness of the model does not quite change when increasing the dataset size, and the accuracy neither changes. Since MNIST is a simple dataset, newly added training examples are likely redundant, and does not increase accuracy nor robustness.

- Generally, a more accurate model has worse robustness. For example, for CIFAR-10, ResNet and DenseNet show much worse robustness than simple models like 2-layer perceptron and 7-layer Simple CNN. This observation is consistent with the findings of Su et al. (2018) on the ImageNet dataset.

Overall, when the models are supplied with small amount of data, increasing training data helps them to find a better and clearer decision boundary, resulted in better accuracy and robustness. However, while more training data may lead to higher accuracy, the decision boundaries can become complicated and delicate, causing adversarial examples to forge. Thus, there is usually a "turning point" where the robustness starts to decrease while the accuracy keeps increasing. The turning point is model and dataset dependent. In our experiments, the turning points occurred in CIFAR-10 and STL-10 due to the abundant data they contain *w.r.t.* Simple CNN, ResNet and DenseNet with data augmentation. For ResNet without data augmentation, the training accuracy has not improved sufficiently due to the lack of data, and therefore the turning point has not reached. Data augmentation essentially increases the size of dataset; consequently models can achieve higher accuracy leading to the emergence of turning points, as a result, decreasing robustness. Similarly the turning point did not occur on Tiny ImageNet due to the data shortage.

## 5 RELATED WORK

**Robustness Evaluation.** One way to evaluate the robustness of a DNN is to find the minimal adversarial distortion (in terms of a particular form of $\ell_p$ norm). A neural network is said to be more robust if it tolerates larger amount of adversarial perturbations. Although finding the exact minimal adversarial distortion is NP-hard (Katz et al., 2017; Sinha et al., 2017; Ehlers, 2017), an adversarial attack can be deemed as an upper bound of the minimal adversarial distortion and is widely adopted for robustness evaluation. Existing work (Goodfellow et al., 2014; Carlini & Wagner, 2016; Xu et al., 2018; Metzen et al., 2017; Xiao et al., 2018b; Sun et al., 2018; Cheng et al., 2018; Carlini & Wagner, 2018; Xiao et al., 2018a; Chen et al., 2018) adopts this technique to evaluate the robustness of DNNs for many different tasks.

On the other hand, several approaches have been proposed to find the lower bounds of minimal adversarial distortion. Szegedy et al. (2013) offers a very loose result by bounding the global Lipschitz constants; Hein & Andriushchenko (2017) provides instance-specific lower bounds by analytically deriving the local Cross-Lipschitz constant for 2-layer networks. Weng et al. (2018b) empirically estimates the local Cross-Lipschitz constant for larger networks without statistical guarantees. Gehr et al. (2018); Weng et al. (2018a) give certified lower bounds of the minimum distortion by exploiting the special structure of ReLU networks. Unfortunately, these approaches can only be applied to relatively small networks due to the restriction of methodology or limited computational resources, therefore providing guaranteed lower bounds for arbitrary networks is still out of reach.

**Accuracy and Robustness Tradeoffs.** Su et al. (2018) study the relationship between accuracy and robustness. The authors investigate 18 well-trained neural networks on ImageNet and compare their accuracy and the robustness. They observe that more accurate models can exhibit worse robustness. For example, according to this study, ResNet is more accurate than AlexNet while being significantly less robust. Schmidt et al. (2018) conclude that robust generalization (*i.e.*, good accuracy and robustness) is significantly harder than standard generalization (*i.e.*, good accuracy); and robust generalization may need significantly more data even under a simple (but unrealistic) assumption that data were drawn from a Gaussian distribution. Our paper sheds light on how training data affect both accuracy and robustness. Specifically, for a given model architecture, we study how accuracy and robustness change with increased amount of training data.

**Data Poisoning Attacks.** Data poisoning attacks generally inject false training data with the aim of corrupting the learned model by malicious users (Steinhardt et al., 2017; Biggio et al., 2012; 2014; Xiao et al., 2015; Rubinstein et al., 2009; Mei & Zhu, 2015; Yang et al., 2017), *etc.*. Under this setting, the accuracy of the learned model drops due to the injected false training data. In our study, we add more natural training data to train a model, the accuracy of the model increases while its robustness may drop.

# 6 CONCLUSION

This paper studies how robustness changes with increased training data for several representative neural networks on different datasets. Our experimental results show that with increased training data, both accuracy and robustness improve initially, however, there exists a turning point after which accuracy keeps increasing while robustness starts to decrease. Such turning points are different for different datasets and neural networks. This study also sheds light on several possible directions for future work such as how to select training data to make a model more robust without sacrificing its accuracy and how to devise new attacks on robustness by flooding a model with excessive amount of natural training data.

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

# 7 APPENDIX

In this section, we show the results of measuring the robustness of models in a different way. For example, if the test accuracy of $\mathcal{M}$ on testing dataset $T$ of size 1000 is 80%, i.e., it has wrong prediction on 200 images in $T$. Then we use the 800 valid inputs as attack images. Here we define an attack is successful on a valid input $x$ if $\mathcal{M}(x') \neq \mathcal{M}(x)$ and $||x - x'||_p \leq \Delta$ where $x'$ is the adversarial image generated by the attack and $\Delta$ is the threshold of the distortions allowed. Intuitively, we say an attack is successful if the model has a wrong prediction on the adversarial image and the distortion of the adversarial image from the original image is within the threshold $\Delta$. Suppose there are 400 successful attacks in this example, then we say the *adversarial accuracy* of $\mathcal{M}$ is $1 - (200 + 400)/1000 = 0.4$. The higher the *adversarial accuracy*, the more robust the model is. Formally, the robustness of the model is measured by

$$Robustness(\mathcal{M}) = 1 - \frac{\text{wrong predictions} + \text{successful attacks}}{|T|} \qquad (5)$$

Note that we cannot compare robustness across different model architectures using this measurement since $\Delta$ for different model architectures varies.

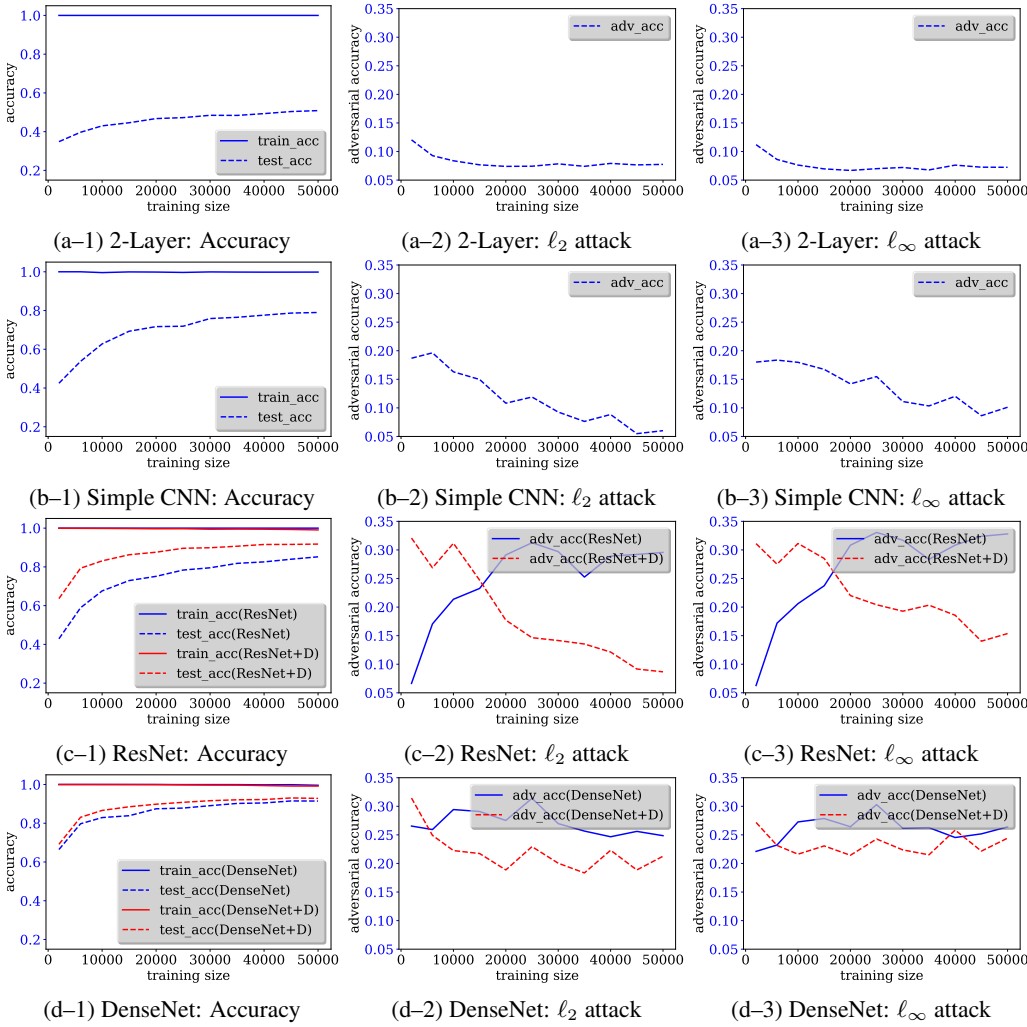

Figure 5: Results of accuracy and robustness of different neural networks on increasing training data on CIFAR-10. (ResNet/DenseNet+D represents ResNet/DenseNet models with data augmentation)

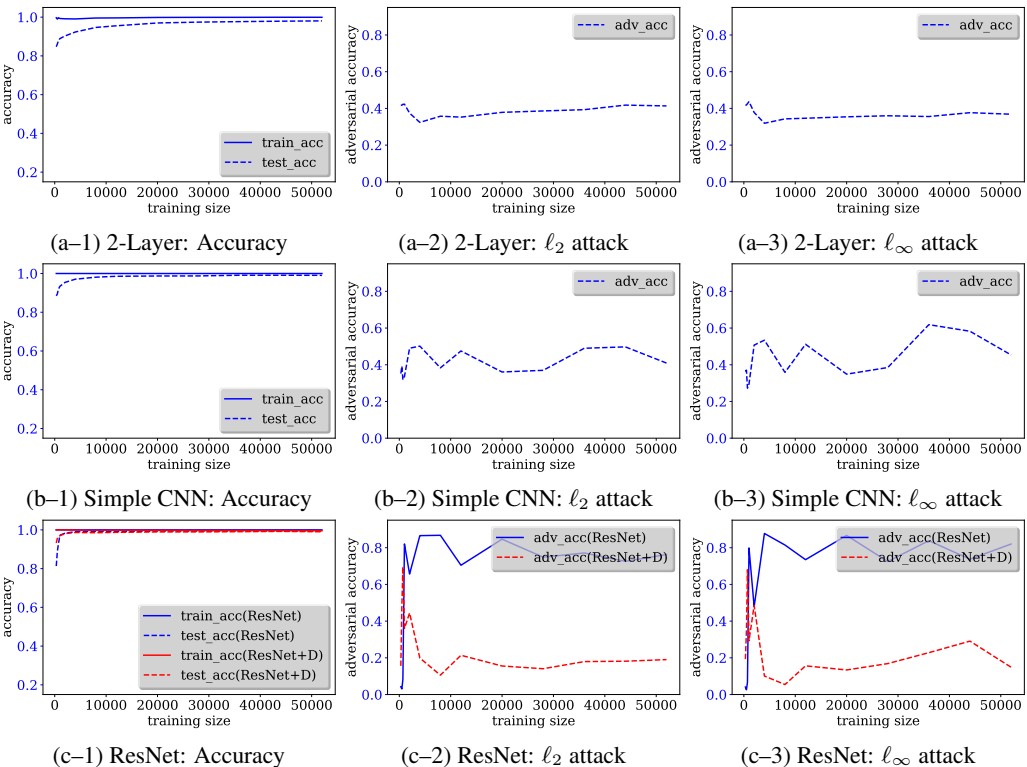

(a–1) 2-Layer: Accuracy     (a–2) 2-Layer: $\ell_2$ attack     (a–3) 2-Layer: $\ell_\infty$ attack

(b–1) Simple CNN: Accuracy     (b–2) Simple CNN: $\ell_2$ attack     (b–3) Simple CNN: $\ell_\infty$ attack

(c–1) ResNet: Accuracy     (c–2) ResNet: $\ell_2$ attack     (c–3) ResNet: $\ell_\infty$ attack

Figure 6: Results of accuracy and robustness of different neural networks on increasing training data on MNIST dataset.(ResNet+D represents ResNet models trained with data augmentation.)

Results on CIFAR-10, MNIST, STL-10 and Tiny ImageNet are shown in Figs 5–8, respectively. The x-axis of each figure shows the size of sub-datasets, and the y-axis shows the accuracy or robustness (measured by adversarial accuracy) of the models trained on each sub-dataset. adv_acc in the figures represents adversarial accuracy. $\ell_2$ attack shows the adversarial accuracy by CW attack, with the threshold on $\ell_2$ distortions. Similarly, $\ell_\infty$ attack shows the adversarial accuracy by I-FGSM attack, with the threshold on $\ell_\infty$ distortions.

Conclusions from this measurement of robustness are consistent with the one used in Section 4.

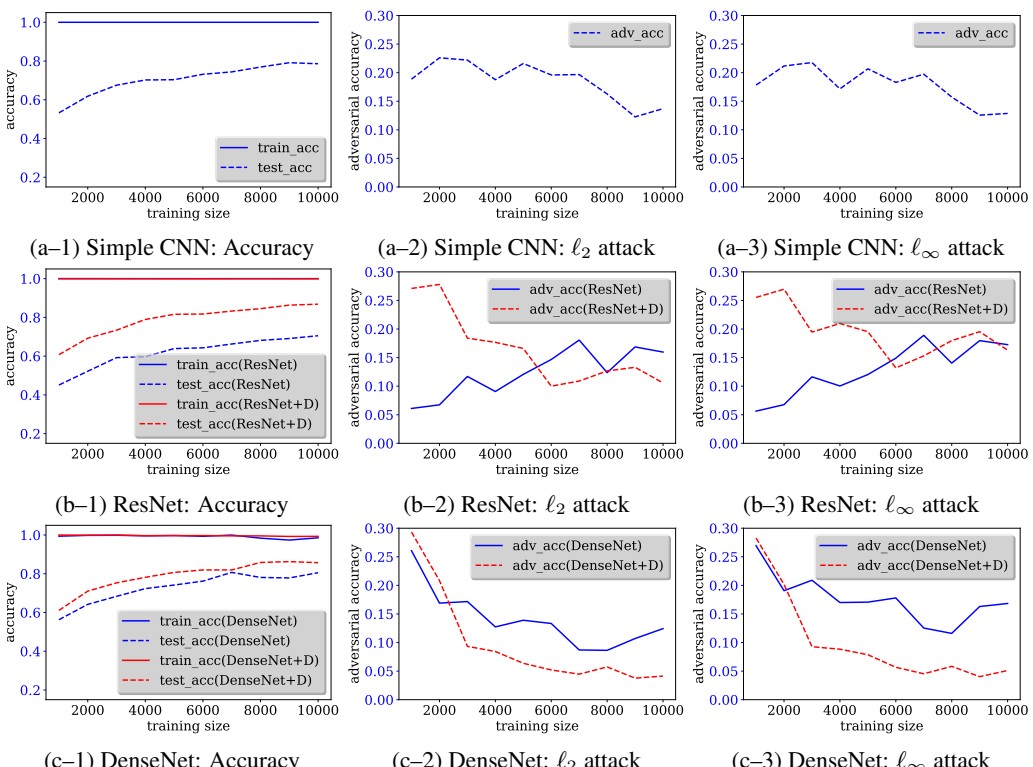

Figure 7: Results of accuracy and robustness of different neural networks on increasing training data on STL-10. (ResNet/DenseNet+D represents ResNet/DenseNet models with data augmentation.)

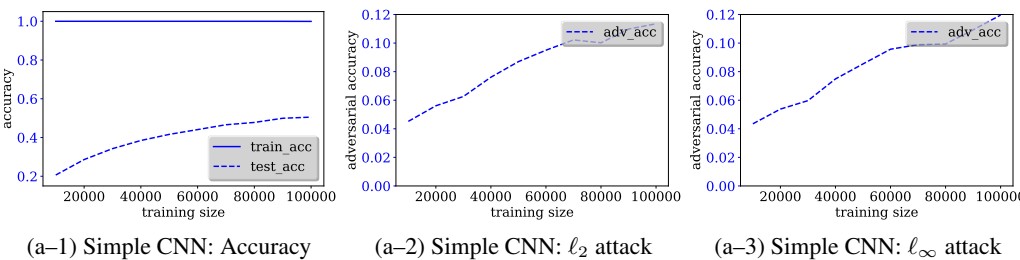

Figure 8: Results of accuracy and robustness of different neural networks on increasing training data on Tiny ImageNet dataset.

