# OpenReview forum: "How Training Data Affect the Accuracy and Robustness of Neural Networks for Image Classification"
_ICLR.cc/2019/Conference_

### Official Review · AnonReviewer3 · 2018-11-02
**Empirical study of variation of accuracy and robustness of networks versus training data size**

**Rating:** 5
**Confidence:** 4

**Review:**

The paper presents an empirical study of how accuracy and robustness vary with increasing training data for four different data sets and CNN architectures. The main conclusion of the study is that while training accuracy generally increases with increasing training data, provided sufficient training data is available for training the network in the first place, the robustness on the other hand does not necessarily increase, and may even decrease.

Similar findings were presented previously in Su et al., 2018. Hence, the current paper contains incremental and marginal new findings versus the existing literature. The paper would also have been a lot stronger and significantly advanced our scientific understanding of the problem if the authors had made some attempt at trying to explain their findings theoretically. In its current form the paper does not contain sufficient contributions for acceptance.

---

> ### Author Response · Authors · 2018-11-24
> **Response to AnonReviewer3**
>
> Thank you for the suggestion to explore how to explain our findings theoretically, which we agree would be interesting.
>
> We wish to clarify the contributions of our work in connection to Su et al., 2018. Compared to Su et al., 2018, we study robustness from a very different perspective --- we study the effect of training data on the robustness of any given model, while Su et al., 2018 study how different model architectures affect robustness. The most important contribution of our work is that, from the experimental results, we have observed that training data can significantly affect the robustness of models. Our findings motivate further work, such as how to further analyze the relationship of robustness and the distribution of training data, and how to enhance the robustness of a model by transforming the training data.

---

> > ### Comment · AnonReviewer3 · 2018-11-27
> > **Response to Authors**
> >
> > I thank the authors for clarifying the contribution of the paper and for providing additional results with other measures of robustness. I have hence revised my rating.

---

### Official Review · AnonReviewer2 · 2018-11-03
**Clear structure and presentation of the empirical evaluation but the significance of the results is not clear.**

**Rating:** 5
**Confidence:** 3

**Review:**

This paper empirically evaluates the effect of the training dataset size on accuracy and robustness against adversarial attacks. The methodology of the paper is generally easy to assess and the overall idea well communicated.

For the motivation example I assume the following assessment holds true. Several linear functions are sampled and compose S_1, S_2, and T. A single linear regression model is used to fit all the data, either S_1 or (S_1 and S_2). If that is the case the experiment is not clear to me since the single linear model can only fit the data mean, mean slope (a) and constant (mu). Since the joint dataset better captures the mean of T the error for the joint training should be lower indeed. However, to actually compare both values the same threshold theta should be used for both and not a percentage of their performance. I would argue that this very simple model does not provide any valuable insight into the problem due to its construction.

The experimental setup presented in Section 4 only considers examples which are classified correctly by all data subsets. However, it is crucial to also consider the mistakes of these subsequent sets. For example, the learned model for the most restrictive dataset is most likely not exposed to a complex decision boundary, therefore it will exhibit a much smoother prediction at the cost that it will simply classify many more examples as the target class. In this case using data perturbations is not even the problem since completely different examples might be classified wrongly. Although not entirely clear, it would be very useful to consider the nearest negative neighbor in the dataset in the embedding space of the classifier to capture this problem at least partially. In general if the test accuracy is lower the learned classifier exhibits less performance, thus, adversary examples, distorted examples are not the main issue since it simply makes mistakes on visually different examples. Therefore, the overall analysis should be much more focused on models which achieve the same test performance but use require less data to achieve this performance.

---

> ### Author Response · Authors · 2018-11-24
> **Response to AnonReviewer2**
>
> Thank you for the helpful comments and suggestions. We will answer the questions as listed:
>
> Q1: For the motivation example I assume the following assessment holds true. Several linear functions are sampled and compose S_1, S_2, and T.
>
> A1: We apologize for the unclear description in the paper.  Only one linear function is used to generate natural training and testing data. In more detail, we generate the data using the function y = a * x + u, where a is a fixed value.  For example, let a = 2, we can sample the data using y = 2x + u. To generate each pair of (x, y), we randomly sample both x and u with x from the range (0, 10) and u from the range (-1, 1).  Then, we compute the corresponding y using the sampled x and u.
>
> Q2: A single linear regression model is used to fit all the data, either S_1 or (S_1 and S_2). If that is the case the experiment is not clear to me since the single linear model can only fit the data mean, mean slope (a) and constant (mu). Since the joint dataset better captures the mean of T the error for the joint training should be lower indeed.
>
> A2: Yes, in the example, the joint dataset better captures the mean of T, and the error for the joint training is lower, i.e., the mean squared error of M2 (computed on more training data) is smaller than that of M1 on T.
>
> Q3: However, to actually compare both values the same threshold theta should be used for both and not a percentage of their performance.
>
> A3: In the example, we do actually use the same threshold \theta for both M1 and M2.  Our use of the symbol \theta might have confused the reviewer, which we apologize --- it is the relative error, not a percentage. In more detail, the robustness of each of the models is measured by the average distortion it can tolerate to make correct predictions for the testing data in T. Here, we say a prediction is correct if the relative error of the prediction and the label is smaller than the given threshold, \theta.  The same threshold is used for M1 and M2. For example, given a testing data t \in T, we compute the robustness of M1 on t as follows. We continue adding bigger and bigger distortions to t by a tiny step.  For each t’, we check if the model still predicts correctly on it (by checking if the relative error of the prediction on t’ and the label exceeds the threshold \theta). If no, we add additional distortions.  If yes, we return |t’ - t| as the robustness value of M1 on t.
>
> Q4: I would argue that this very simple model does not provide any valuable insight into the problem due to its construction.
>
> A4: Through this simple example, we want to show analytically (rather than empirically) the existence of our observed phenomenon, i.e., models trained on more data can be more accurate but less robust.  In the example, we generate two datasets, S1 and S2 with S1 \subset S2. Next, we compute two linear regression models M1 and M2 on S1 and S2 respectively based on a closed-form calculation. Then, we show that M2 is more accurate than M1 since the mean squared error of M2 on the testing set T is smaller than that of M1. And for each testing data t in T, we show that M1 tolerates more distortions on t to still make correct predictions than M2, which indicates that M2 is less robust than M1. The phenomenon, i.e, with more training data, the linear regression model can be more accurate but less robust, motivated us to conduct this extensive empirical analysis to investigate and understand if (and how much) this phenomenon holds for realistic neural network models for image classification tasks.
>
> The reviewer’s last question concerns how to measure and compare the robustness of models. In this paper, we use the average distortions on the commonly correctly predicted images to compare the robustness of models. We add further results in the appendix using a different metric (including the mistakes of the models), which confirm the same pattern. Usually the test accuracy increases with more training data. It would be interesting future research to study how to select training data to maintain test accuracy, but deteriorate robustness.

---

### Official Review · AnonReviewer4 · 2018-11-14
**An empirical study of the influence of training data size on model robustness**

**Rating:** 4
**Confidence:** 4

**Review:**

This paper conducts an empirical analysis of the effect of training data size on the model robustness to adversarial examples. The authors compared four different NN architectures using four different datasets for the task of image classification. Overall, the paper is easy to follow and clearly written.

However, since Su et al., 2018, already presented similar findings, I do not see any major contribution in this paper. Additionally, I would expect the authors to conduct some more analysis of their results besides acc. and distortion levels. For examples, investigate the type of mistakes the models have made, compare models with the same test acc. but different amount of training data used to get there, some analysis/experiments to explain these findings (monitor models parameters/grads during training, etc.)

---

> ### Author Response · Authors · 2018-11-24
> **Response to AnonReviewer4**
>
> Thank you for your efforts reviewing our paper and providing helpful comments and suggestions.
>
> First, since the question about the contribution is the same as the one raised by Reviewer 3, we repeat our answer here.  We wish to clarify the contributions of our work in connection to Su et al., 2018.  Compared to Su et al., 2018, we study robustness from a very different perspective --- we study the effect of training data on the robustness of any given model, while Su et al., 2018 study how different model architectures affect robustness. The most important contribution of our work is that, from the experimental results, we have observed that training data can significantly affect the robustness of models. Our findings motivate further work, such as how to further analyze the relationship of robustness and the distribution of training data, and how to enhance the robustness of a model by transforming the training data.
>
> Second, we add in the appendix results on comparing the robustness of models in a different way by taking the mistakes of models into consideration. From our experimental results, the test accuracy usually increases with more training data. The focus of our current study is how the accuracy and robustness of models change with respect to increased training data (by randomly partitioning the original training dataset into S1 \subset S2,...\subset Sn, all of which are balanced). As we commented in our response to one comment in Review 2, it would be interesting future research to study how to select training data to maintain test accuracy, but deteriorate robustness.

---

### Author Response · Authors · 2018-11-24
**Revision Summary**

We thank the reviewers for the helpful feedback, which has helped us refine our paper and guided us to clarify some important misinterpretations in the reviews, in particular, concerning the contributions of our work with respect to Su et al., 2018.  We have updated our paper accordingly. A major update is the extra appendix to provide a different measurement of robustness based on questions from the reviewers.

---

### Meta-Review · Area_Chair1 · 2018-12-17
**reject**

**Confidence:** 5
**Recommendation:** Reject

**Metareview:**

The reviewers conclude the paper does not bring an important contribution compared to existing work. The experimental study can also be improved.